# Risk Factors for Bleeding Events in Japanese Patients with Advanced Lung Cancer: Data from the Rising-VTE/NEJ037 Study

**DOI:** 10.3390/cancers16020301

**Published:** 2024-01-10

**Authors:** Keita Kawakado, Yukari Tsubata, Takamasa Hotta, Masahiro Yamasaki, Nobuhisa Ishikawa, Kazunori Fujitaka, Tetsuya Kubota, Kunihiko Kobayashi, Takeshi Isobe

**Affiliations:** 1Department of Internal Medicine, Division of Medical Oncology and Respiratory Medicine, Shimane University Faculty of Medicine, 89-1 Enya-cho, Izumo 693-8501, Japan; keita.kawakado1112@med.shimane-u.ac.jp (K.K.); takamasa@med.shimane-u.ac.jp (T.H.); isobeti@med.shimane-u.ac.jp (T.I.); 2Department of Respiratory Disease, Hiroshima Red Cross Hospital and Atomic-Bomb Survivors Hospital, 1-9-6, Senda-Machi, Naka-ku, Hiroshima 730-8619, Japan; myamasanjp@yahoo.co.jp; 3Department of Respiratory Medicine, Hiroshima Prefectural Hospital, 1-5-54 Ujina-Kanda, Minami-ku, Hiroshima 734-8530, Japan; nobuhisa_9@msn.com; 4Department of Respiratory Medicine, Hiroshima University Hospital, 1-2-3 Kasumi, Minami-ku, Hiroshima 734-8553, Japan; fujikazu@hiroshima-u.ac.jp; 5Department of Respiratory Medicine and Allergology, Kochi University Hospital, 185-1 Kohasu, Oko-Cho, Nankoku 783-8505, Japan; kubotat@kochi-u.ac.jp; 6Department of Respiratory Medicine, Saitama Medical University International Medical Center, 1397-1 Yamane, Hidaka 350-1298, Japan; danbe@outlook.jp

**Keywords:** lung cancer, venous thrombosis, bleeding, edoxaban

## Abstract

**Simple Summary:**

Cancer increases the risk of venous thrombosis (VTE) and anticoagulant therapy is often used to treat this condition. Therefore, it is essential to identify risk factors for bleeding events in these patients. In our study, we used data from the Rising-VTE/NEJ037 study, which involved Japanese patients with advanced lung cancer, to calculate these risk factors. We found that complications with VTE were the most significant risk factor. This finding underscores the importance of carefully assessing the risks and benefits before starting anticoagulant therapy in patients with cancer. Our research aims to inform safer treatment strategies for patients with advanced lung cancer, which could have meaningful implications for their care and treatment approaches in the research community.

**Abstract:**

Despite the occurrence of various hemorrhagic events during advanced lung cancer treatment, few researchers have reported on their risk factors. Moreover, the development of cancer-related thromboembolism indicates anticoagulant use. However, adverse events such as bleeding should be monitored. In this study, we aimed to identify factors that influence the onset of hemorrhagic events in patients with lung cancer. The Rising-VTE/NEJ037 study was a multicenter, prospective, observational study. A total of 1008 patients with lung cancer who were unsuitable for radical resection or radiation were enrolled and followed up for 2 years. Multivariate analysis using a Cox proportional hazard model was performed to compare the outcomes of the time to the onset of hemorrhagic events for 2 years after registration. Hemorrhagic events occurred in 115 patients (11.4%), with 35 (30.4%) experiencing major bleeding. Significant risk factors included venous thromboembolism (VTE) (hazard ratio [HR]: 4.003, *p* < 0.001) and an Eastern Cooperative Oncology Group Performance Status score of 1 (HR: 2.476, *p* < 0.001). Factors that significantly reduced hemorrhagic event risk were female sex (HR: 0.454, *p* = 0.002) and M1a status (HR: 0.542, *p* = 0.038). VTE is a risk factor for hemorrhagic events in patients with advanced lung cancer, and risks associated with anticoagulant therapy should be considered.

## 1. Introduction

Venous thromboembolism (VTE) is a common medical complication in patients with cancer, and the risk of VTE development is 4–20 times greater in patients with cancer than in those without cancer [1,2,3,4,5]. In addition, lung cancer has a high risk of VTE [2]. The incidence of VTE increases with chemotherapy [6,7]. There are also reports that active cancers are associated with VTE in approximately 20% of cases [8,9]. In addition, there are reports that cancer patients with VTE have a worse prognosis and higher mortality rate than those without VTE [7,10]. Anticoagulation therapy is important in the treatment of VTE, but it can cause bleeding side effects, and individual bleeding risks must be considered when using this therapy. Treating venous thromboembolism in patients with cancer is particularly challenging due to the increased risks of both recurrent thrombosis and bleeding compared to patients without cancer [11]. Studies have shown the effectiveness of low-molecular-weight heparin and vitamin K antagonists in managing cancer-associated VTE [11,12,13,14,15]. Additionally, direct oral anticoagulant agents have been reported to be as effective as vitamin K antagonists for treating VTE, with a lower risk of bleeding events [16,17,18]. In the Hokusai VTE cancer trial, of the 522 patients treated with edoxaban for cancer-related VTE, 36 (6.9%) had major bleeding, and 76 (14.6%) had clinically relevant non-major bleeding [19]. We conducted the Rising-VTE/NEJ037 study, a multicenter prospective observational study on patients with lung cancer, and 2 years after enrollment, 9.9% of patients had developed VTE, of which 75% were asymptomatic [20]. The dataset of the Rising-VTE/NEJ037 study was used to analyze the bleeding risk in Japanese patients with advanced lung cancer. In addition, the HAS-BLED and VTE-BLEED scores are known to be indicators of bleeding risk [21,22]. HAS-BLED and VTE-BLEED scores were also calculated to compare existing scoring for bleeding events and identify risk factors in this study.

## 2. Materials and Methods

This study was conducted in accordance with the principles of the Declaration of Helsinki and Good Clinical Practice Guidelines. The study protocol was approved by the Shimane University Institutional Review Board, based on the Clinical Trials Act enacted in Japan in 2017 and published in the Japan Registry of Clinical Trials (jRCTs061180025). Written informed consent was obtained from all patients.

### 2.1. Patients

Data were collected and examined as part of a multicenter prospective observational study in Japan. The enrolment period of the study was from June 2016 to August 2018. The main enrollment criteria were as follows: patients aged 20 years or older, those with lung cancer for which radical resection or radiation therapy was difficult, those with Eastern Cooperative Oncology Group Performance Status (ECOG PS) scores of 0–3, and those expected to survive for six months or longer. There were no exclusion criteria for case enrollment. Edoxaban, a direct oral anticoagulant, was administered to patients diagnosed with VTE at enrollment [20]. Exclusion criteria for edoxaban administration were patients with hypersensitivity to edoxaban, decreased renal function (creatinine clearance <30 mL/min), liver disease with coagulation abnormalities, and those who have already undergone surgical treatment or treatment for recently diagnosed VTE. If the exclusion criteria for edoxaban administration were met for patients with VTE, anticoagulation therapy was administered at the discretion of the attending physician.

### 2.2. Bleeding Events

Bleeding events were assessed according to the International Society on Thrombosis and Hemostasis criteria [23]. Clinically evident bleeding that met at least one of the following conditions was considered major bleeding: decrease in hemoglobin level by ≥2 g/dL, transfusion of ≥2 units (500 mL/unit) of packed red blood cells or whole blood, bleeding in critical areas (intracranial bleeding, intraspinal bleeding, intraocular bleeding, pericardial bleeding, intra-articular bleeding, intramuscular bleeding accompanied by compartment syndrome, and retroperitoneal bleeding), and fatal bleeding. Meanwhile, bleeding that did not meet the criteria for major bleeding but was deemed clinically important at the discretion of the attending physician, for example, gastrointestinal bleeding or hemoptysis that does not meet the definition of major bleeding, was considered clinically relevant non-major bleeding. If a bleeding event occurred, treatment was performed at the discretion of each attending physician.

### 2.3. Hypertension, Abnormal Liver/Renal Function, Stroke History, Bleeding History or Predisposition, Labile INR, Elderly, Drugs/Alcohol Concomitantly (HAS-BLED) Score

For patients with atrial fibrillation, the HAS-BLED score is known to predict the risk of bleeding side effects during the oral administration of anticoagulants [21]. Abnormal renal/liver function, risk of stroke, bleeding history or predisposition, labile international normalized ratio, old age (>65 years), and drug/alcohol use are used to predict the risk of developing a major bleeding event, with a maximum of nine points. The incidence rates of major bleeding events for these factors were 1.1%, 1.0%, 1.9%, 3.7%, 8.7%, and 12.5%, respectively (range: 0–5 points), and patients with a score of ≥3 were regarded as having a high risk for major bleeding events. To compare the risk of bleeding due to edoxaban and that associated with lung cancer, we used the HAS-BLED score, which has been validated in patients with atrial fibrillation. The history of HAS-BLED events or predisposition, labile international normalized ratio, age (>65 years), and drug/alcohol use were calculated using data from the Rising-VTE/NEJ037 study. In this study, for the edoxaban group, we were unable to calculate the history of alcohol. Therefore, alcohol history was calculated as 1 point and 0 points (Figure 1).

### 2.4. Venous Thromboembolic Disease and Bleeding (VTE-BLEED) Score

According to the VTE-BLEED score, all the patients fell under the high-risk group because all cases registered in this study were active cancers [22]. Therefore, only the HAS-BLED scores were used in this study.

### 2.5. Statistical Analyses

Data from the Rising-VTE/NEJ037 dataset were subjected to multivariate analysis using the Cox proportional hazards model with “the time to onset of a bleeding event” as the outcome. All statistical analyses were performed using SPSS Statistics version 24.0 (IBM Japan, Ltd., Tokyo, Japan). We used the Cox proportional hazards model to examine the risk of bleeding in Japanese patients with advanced lung cancer. *p* values < 0.05 were considered to be statistically significant.

## 3. Results

### 3.1. Patients’ Characteristics and Bleeding Events

A total of 1021 patients were enrolled from 35 Japanese hospitals, and 13 were excluded due to missing data. Finally, 1008 patients were included. The median age of the enrolled patients was 70 (range, 30–94) years, and most were male (714 patients, 70.8%) and had good ECOG PS (0–1, 80.6%). The most common histological subtype of lung cancer was adenocarcinoma (641 patients, 63.6%). Disease stage was assessed according to the Union for International Cancer Control, 7th edition, tumor-node-metastasis staging system for lung cancer [24]; M1a, b stage IV disease accounted for 80% of all cases [11]. At the initial diagnosis, 62 patients had VTE, of which 44 were treated with edoxaban (Figure 2). Further, 115 (11.4%) had bleeding events: 80 were non-major bleeding events, and 35 were major bleeding events. In the group that had VTE and received edoxaban treatment (44 patients), bleeding events occurred in 15 patients, with major bleeding events in four. Of the 964 patients who did not receive edoxaban, bleeding events occurred in 100 (10.4%). Major bleeding events occurred in 31 patients (Figure 3).

Patient characteristics between the groups with and without bleeding/hemorrhagic events (Table 1). In the group with bleeding events, the proportion of males was slightly greater, proportion of patients with ECOG PS 1 score was higher, and proportion of patients with squamous cell carcinoma was higher. Additionally, complications of atrial fibrillation and history of anticoagulant use were slightly more common in the group with bleeding events.

### 3.2. Bleeding Site

The most common site of bleeding was the respiratory system, as observed in 42 of 115 patients. The gastrointestinal system was the next most frequent site, followed by the nose. The distribution of bleeding sites was uniform in the edoxaban group (Figure 4).

### 3.3. Multivariate Analysis

We performed multivariate analysis using the Cox proportional hazards model with the time to the onset of a bleeding event as the outcome (Table 2). The factors that significantly increased the risk of hemorrhagic events were VTE and an ECOG PS of 1. The factors that significantly reduced the risk of hemorrhagic events were female sex and pleural dissemination (M1a). In particular, among patients with VTE, the hazard ratio for hemorrhagic events was as high as 4.003.

### 3.4. HAS-BLED Score

Edoxaban was administered only to patients with no previous history of anticoagulant therapy. As the treatment of atrial fibrillation included anticoagulant therapy, it can be inferred that no patients in the edoxaban group had atrial fibrillation.

When drinking history was set at 0 points, the HAS-BLED score was 0 points for 10 patients, 1 point for 21 patients, 2 points for 12 patients, and 3 points for one patient. Only one patient in the edoxaban group had a HAS-BLED score ≥3 points. Major bleeding events occurred in two patients at 0 points, one patient at 1 point, and one patient at 2 points (Figure 5). When drinking history was set at 1 point, the HAS-BLED score was 1 point for 10 patients, 2 points for 21 patients, 3 points for 12 patients, and 4 points for one patient. There was only one patient with a HAS-BLED score of ≥4 points in the edoxaban group. Major bleeding events occurred in two patients at 1 point, one patient at 2 points, and one patient at 3 points (Figure 6).

## 4. Discussion

The Rising-VTE/NEJ037 study is one of the largest prospective observational studies of venous thrombosis and hemorrhagic events in Japanese patients with advanced lung cancer. Using the results of the Rising-VTE/NEJ037 study, we performed a post hoc analysis to evaluate the risk factors for bleeding events in Japanese patients with advanced lung cancer. To our knowledge, our study is the first to analyze the risk factors for bleeding events in patients with advanced lung cancer using large-scale data from a prospective observational study. In the Rising-VTE/NEJ037 study, the median overall survival was 24.0 months (95% confidence interval [CI]: 16.8–not estimable) in the group complicated by VTE and treated with edoxaban and 19.2 months (95% CI: 16.8–21.6) in the observation group, indicating no significant difference in survival time (*p* = 0.793) among the two groups [20]. This indicates that VTE treatment may contribute to improving the prognosis of patients with advanced lung cancer; it is crucial to consider the risks versus benefits of bleeding events and to administer the treatment with due caution.

Studies have shown that cancer patients are more prone to bleeding events compared to non-cancer patients [25,26,27]. Identifying risk factors for these bleeding events in cancer patients is paramount. VTE complications were found to be the largest risk factor for bleeding events in Japanese patients with advanced lung cancer. In addition, female sex and M1a status were associated with a reduction in bleeding events. Further, 44 of the 62 patients with VTE were taking edoxaban, which may have been a significant risk factor for bleeding events.

A high bleeding rate was observed in the group with a HAS-BLED score of ≤2, which was a low-risk group, suggesting that VTE complicating advanced lung cancer is a bleeding risk factor. Active cancer was reported to be a major predictor of recurrent VTE and bleeding [28]. A previous report examined the incidence of VTE and risk factors for bleeding events across cancer types and reported that the VTE complication rate for patients with solid tumors was 5.9% and that VTE complication was a risk factor for increased bleeding events in the multivariate analysis, similar to the results of our study (hazard ratio [HR], 2.39; 95% confidence interval [CI], 1.32–4.31; *p* = 0.004) [28]. Lung cancer was also listed as a risk factor for increased bleeding events (HR, 1.77; 95% CI, 1.01–3.11; *p* = 0.046) [29]. Cancer and VTE are diseases that are presumed to have already developed coagulation system abnormalities, which are expected to increase the risk of bleeding; however, the detailed underlying mechanism remains unknown. ECOG PS 1 is another factor that increases the risk of bleeding, and we considered the possibility that more bleeding events occurred in the symptomatic patient population than in patients with ECOG PS 0. Furthermore, in this study, no patients had platelet counts <20,000 at the time of enrollment, and no cases required platelet transfusions owing to chemotherapy. Therefore, it was considered that low platelets were unlikely to be a risk factor for bleeding events.

Conversely, the factors that reduced bleeding events were female sex and pleural dissemination (M1a), possibly because of the lower prevalence of smoking in women. Regarding the group with pleural dissemination (M1a), it is thought that the risk of bleeding events was reduced because distant metastasis was not likely to be involved in the pulmonary/bronchial expenditure of blood, which is a concern for bleeding events in lung cancer.

### Limitations

This study had some limitations. First, this study investigated the risk factors for bleeding events in Japanese patients with advanced lung cancer; however, the underlying mechanisms are unknown. Therefore, we are currently conducting studies to elucidate the mechanisms involved in thrombus formation and bleeding. Second, this bleeding risk was internally validated in patients enrolled in the Rising-VTE/NEJ037 study, however, external validity was not demonstrated. Third, all patients enrolled in this study were Japanese, and whether this risk factor is the same for other ethnicities is unclear. In addition, it is known that Japanese people generally have a high rate of driver gene mutations for lung cancer; therefore, the bleeding tendencies of Japanese patients may be different from those of patients of other ethnicities. Fourth, a large proportion of patients with VTE were treated with edoxaban, and we strongly believe that VTE complications mostly contributed to the bleeding events, although edoxaban may also have contributed. Fifth, although there has been research on the pharmacokinetics of edoxaban in EGFR gene mutation-positive patients using some of the patients enrolled in the Rising-VTE/NEJ037 study [30], measuring pharmacokinetics and blood levels in all patients was not possible. Future large-scale, multi-center, longitudinal studies with diverse population samples should be conducted in real-world settings and with standardized scales to confirm the generalizability of our findings and to add to the evidence database to extract valuable results. We are currently preparing a prospective study for external validation; we also aim to evaluate the risk factors for bleeding events.

## 5. Conclusions

In Japanese patients with advanced lung cancer, complications with VTE were observed to increase the risk of bleeding the most. Anticoagulant therapy is sometimes performed for patients with advanced lung cancer complicated by VTE; however, the indication for treatment must be determined after fully considering the risk of bleeding. In the future, we would like to consider creating a scoring system for bleeding risk when using anticoagulant therapy for patients with advanced lung cancer.

## Figures and Tables

**Figure 1 cancers-16-00301-f001:**
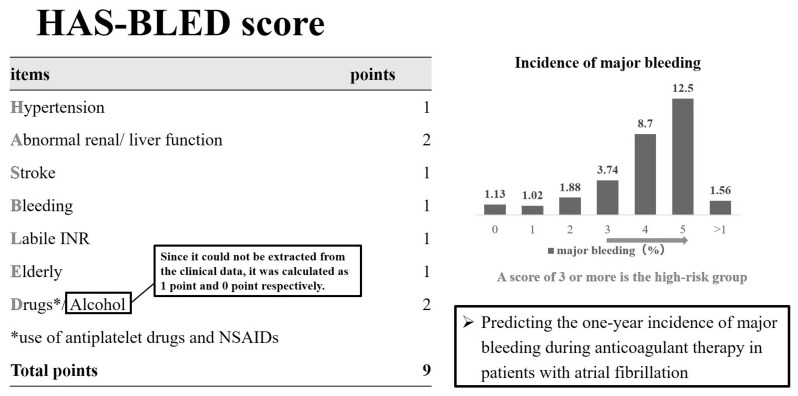
Hypertension, abnormal liver/renal function, stroke history, bleeding (HAS-BLED) score.

**Figure 2 cancers-16-00301-f002:**
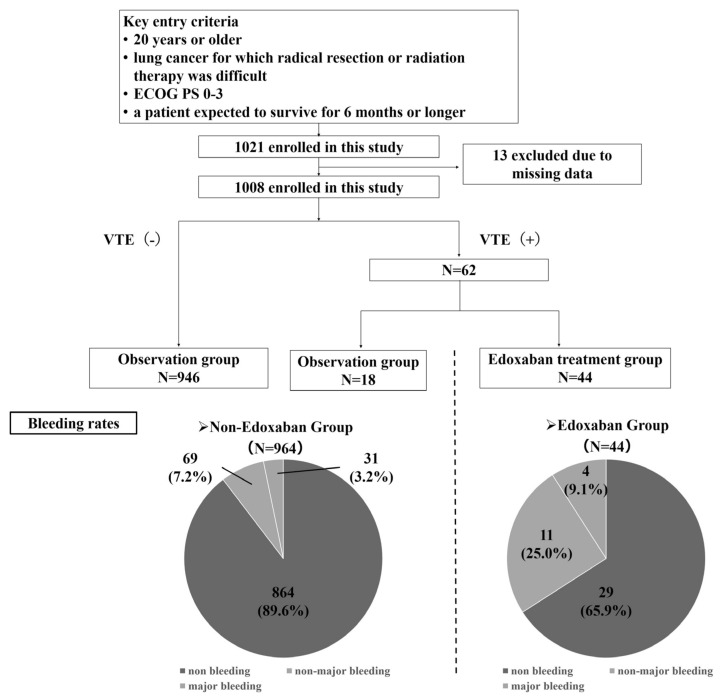
Cluster classification for venous thromboembolism (VTE) screening.

**Figure 3 cancers-16-00301-f003:**
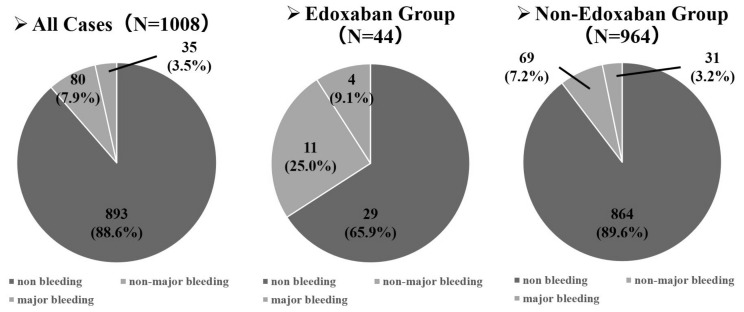
The rates of bleeding events and major bleeding events.

**Figure 4 cancers-16-00301-f004:**
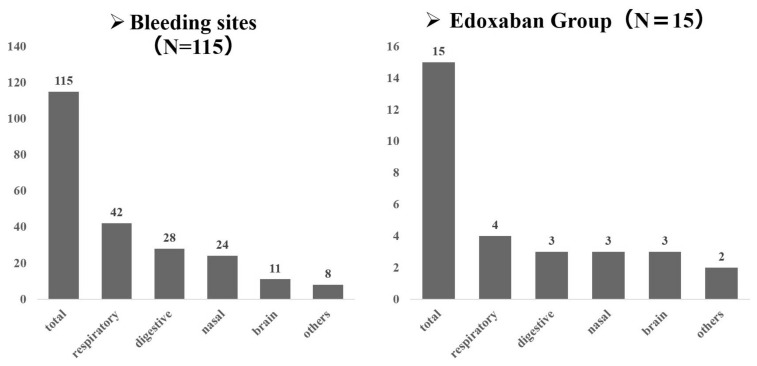
The distribution of bleeding sites.

**Figure 5 cancers-16-00301-f005:**
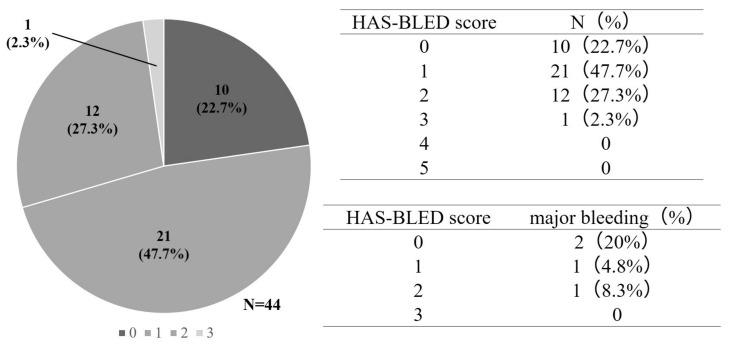
Hypertension, abnormal liver/renal function, stroke history, bleeding history or predisposition, labile INR, elderly, drugs/alcohol concomitantly.

**Figure 6 cancers-16-00301-f006:**
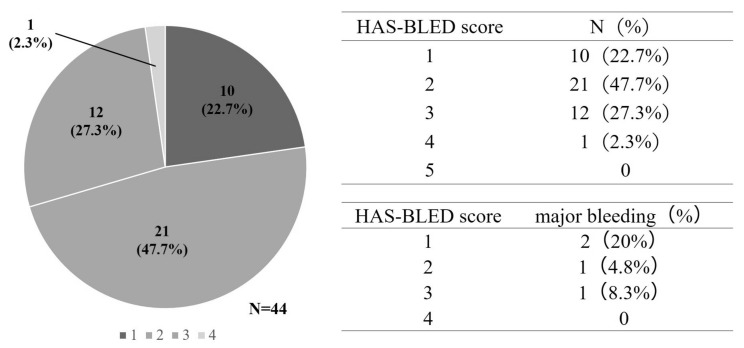
Hypertension, abnormal liver/renal function, stroke history, bleeding (HAS-BLED) scores when drinking history is set at 0 point (A) and when drinking history is set at 1 point.

**Table 1 cancers-16-00301-t001:** Patient characteristics in groups with and without bleeding/hemorrhagic events.

All *n* = 1008	All *n* = 1008	With Hemorrhagic Events *n* = 115	Without Hemorrhagic Events *n* = 893
Age [years]			
Median	70	71	71
Range	30–94	44–88	30–94
Sex [%]			
Male	714 [70.8]	96 [83.5]	618 [69.2]
Female	294 [29.2]	19 [16.5]	275 [30.8]
ECOG PS [%]			
0	403 [40.0]	27 [23.5]	376 [42.1]
1	490 [40.6]	77 [67.0]	413 [46.2]
2	74 [7.3]	6 [5.2]	68 [7.7]
3	41 [4.1]	5 [4.3]	36 [4.0]
Histological type [%]			
Adenocarcinoma	641 [63.6]	65 [56.5]	576 [64.5]
Squamous	187 [18.6]	29 [25.2]	158 [17.7]
Small cell	137 [13.6]	14 [12.2]	123 [13.8]
Others	43 [4.3]	7 [6.1]	36 [4.0]
Clinical stage [%]			
T factor			
T1	160 [16.8]	15 [13.0]	145 [16.2]
T2	255 [26.8]	29 [25.2]	226 [25.3]
T3	213 [22.4]	26 [22.6]	187 [20.9]
T4	287 [30.1]	32 [27.8]	255 [28.6]
Tx	37 [3.9]	2 [1.7]	35 [3.9]
Missing	56	11 [9.6]	45 [5.0]
N factor			
N0	195 [20.2]	20 [17.4]	175 [19.6]
N1	98 [10.2]	8 [7.0]	90 [10.1]
N2	268 [27.8]	32 [27.8]	236 [26.4]
N3	402 [41.7]	47 [40.9]	355 [39.8]
Missing	45	8 [7.0]	37 [4.1]
M factor			
M0	192 [20.0]	29 [25.2]	163 [18.3]
M1a	228 [23.8]	19 [16.5]	209 [23.4]
M1b	540 [56.3]	59 [51.3]	481 [53.9]
Missing	48	8 [7.0]	40 [4.5]
Complications of atrial fibrillation [%]	32 [3.2]	7 [6.1]	25 [2.8]
Complications of vascular disease ^1^ [%]	88 [8.7]	10 [8.7]	78 [8.7]
History of anticoagulant use [%]	59 [5.9]	11 [9.6]	48 [5.4]

^1^ vascular disease includes cerebral infarction and myocardial infarction.

**Table 2 cancers-16-00301-t002:** Results of the multivariate analysis of factors influencing bleeding events in patients with lung cancer.

	N	HR	95% CI	*p* Value
VTE (+) [vs. VTE (−)]	100	4.003	2.470–6.488	<0.001
Female (vs. male)	294	0.454	0.277–0.742	0.002
NSCLC (vs. SCLC)	871	0.867	0.496–1.516	0.616
Adenocarcinoma (vs. SCLC)	641	0.731	0.506–1.058	0.096
Squamous (vs. SCLC)	187	1.522	0.999–2.318	0.051
Others (vs. SCLC)	43	1.323	0.582–3.010	0.504
T type				
T1a	28	1.000	Reference	
T1b	132	0.574	0.183–1.801	0.341
T2a	195	0.866	0.301–2.497	0.790
T2b	60	0.569	0.153–2.120	0.401
T3	213	0.848	0.296–2.430	0.759
T4	287	0.787	0.278–2.224	0.651
TX	37	0.372	0.068–2.029	0.253
N type				
N0	195	1.000	Reference	
N1	98	0.789	0.347–1.790	0.570
N2	268	1.170	0.669–2.045	0.582
N3	402	1.156	0.685–1.951	0.587
M type				
M0	192	1.000	Reference	
M1a	228	0.542	0.304–0.966	0.038
M1b	540	0.732	0.469–1.142	0.169
ECOG PS				
0	403	1.000	Reference	
1	490	2.476	1.597–3.838	0.000
2	74	1.249	0.516–3.024	0.623
3	41	1.996	0.769–5.183	0.156
NSCLC Stage				
Ⅰa	0	-	-	-
Ⅰb	2	5.23	0.701–39.007	0.107
Ⅱa	4	1.795	0.241–13.380	0.568
Ⅱb	4	1.671	0.224–12.451	0.616
Ⅲa	28	1.198	0.450–3.192	0.718
Ⅲb	73	1.248	0.621–2.508	0.535
Ⅳ	623	0.657	0.396–1.090	0.104
Postoperative recurrence	137	1.000	Reference	

HR: hazard ratio, CI: confidence interval, NSCLC: non-small cell lung cancer, SCLC: small cell lung cancer.

## Data Availability

Data availability is restricted due to privacy and ethical restrictions.

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
