# Peer review of "Risk Factors for Bleeding Events in Japanese Patients with Advanced Lung Cancer: Data from the Rising-VTE/NEJ037 Study"

_cancers, 2024, doi:10.3390/cancers16020301_

Round 1
Reviewer 1 Report
Comments and Suggestions for Authors
The topic is interesting and the paper is quite well written. Nevertheless, in my opinion, some parts need to be improved, I have some comments:
1) Factors affecting hemorrhagic events in patients with advanced 2 lung cancer: from the Rising-VTE/NEJ037 study. It is recommended to improve the title to clarify the aim of the study and delete the acronym
2) Abstract. Hemorrhagic events occurred in 115 patients 37 (11.4%), 35 (30.4%) of whom had major bleeding events. Factors that significantly increased the risk 38 of hemorrhagic events included venous thromboembolism (VTE) and an Eastern Cooperative On- 39 cology Group Performance Status score of 1. Factors that significantly reduced the risk of hemor- 40 rhagic events were female sex and M1a status. VTE is a risk factor for hemorrhagic events in patients 41 with advanced lung cancer, and risks associated with anticoagulant therapy should be considered. Thi part might be beneficial to include a sentence in the abstract that briefly summarizes the key findings of the study. This can provide readers with a quick overview of the research.
3) The data set 63 of the Rising-VTE/NEJ037 study was used to analyze the bleeding risk in Japanese patients 64 with advanced lung cancer. In addition, the HAS-BLED and VTE-BLEED scores are 65 known to be indicators of bleeding risk [22,23]. In this study, we also examined the bleed- 66 ing risk using an existing scoring system. Please improve the description of this part and underline the novelty of the study.
4) 2.5. Statistical analyses 118 Data from the Rising-VTE/NEJ037 dataset were subjected to multivariate analysis us- 119 ing the Cox proportional hazards model with “the time to onset of a bleeding event” as 120 the outcome. All statistical analyses were performed using SPSS Statistics version 24.0 121 (IBM Japan, Ltd., Tokyo, Japan). We performed multivariate analysis using the Cox pro- 122 portional hazards model with the time to onset of a bleeding event as the outcome. P val- 123 ues <0.05 were considered to be statistically significant. Please, improve this part and underline the statistical tests used to evaluate the data.
5) Figure 2 and 3. If the images can be higher resolution that would be better for the reader.
6) 4. Discussion 193 The Rising-VTE/NEJ037 study is one of the largest prospective observational studies 194 of venous thrombosis and hemorrhagic events in Japanese patients with advanced lung 195 cancer. To our knowledge, our study is the first to analyze the risk factors for bleeding 196 events in patients with advanced lung cancer using large-scale data from a prospective 197 observational study. In the Rising-VTE/NEJ037 study, the median overall survival was 198 24.0 months (95% confidence interval [CI]: 16.8–not estimable) in the VTE+ve and edoxa- 199 ban treatment group and 19.2 months (95% CI: 16.8–21.6) in the observation (VTE –ve, no 200 edoxaban) group, indicating no significant difference (p=0.793) [21]. This indicates that 201 VTE treatment may contribute to improving the prognosis of patients with advanced lung 202 cancer; it is crucial to consider the risks versus benefits of bleeding events and to admin- 203 ister the treatment with due caution. The discussion section needs to be improved. It could be interesting to record the aim of the study. It is necessary to be more concise in the presentation of the facts, clarifying the results obtained and comparing them with previous or similar studies. However, it is interesting to answer the questions that arise from these results, backed up by published literature.
7) 5. Conclusions 254 VTE is a risk factor for hemorrhagic events in advanced lung cancer, and it is neces- 255 sary to consider the risks associated with the administration of anticoagulant therapy. The conclusions might be beneficial to include a sentence in the last part that briefly summarizes the key findings of the study. This can provide readers with a quick overview of the research.
Comments on the Quality of English Language
Minor changes of English language are required
Author Response
1) Factors affecting hemorrhagic events in patients with advanced lung cancer: from the Rising-VTE/NEJ037 study. It is recommended to improve the title to clarify the aim of the study and delete the acronym
Response: Thank you for your comment. As you pointed out, we have changed the title to better reflect the content as follows: “Risk factors for bleeding events in Japanese patients with advanced lung cancer: data from the Rising-VTE/NEJ037 study”
2) Abstract. Hemorrhagic events occurred in 115 patients 37 (11.4%), 35 (30.4%) of whom had major bleeding events. Factors that significantly increased the risk of hemorrhagic events included venous thromboembolism (VTE) and an Eastern Cooperative Oncology Group Performance Status score of 1. Factors that significantly reduced the risk of hemorrhagic events were female sex and M1a status. VTE is a risk factor for hemorrhagic events in patients with advanced lung cancer, and risks associated with anticoagulant therapy should be considered. This part might be beneficial to include a sentence in the abstract that briefly summarizes the key findings of the study. This can provide readers with a quick overview of the research.
Response: Thank you for your comment. As you pointed out, we have briefly added the main results in the abstract as follows:
“Hemorrhagic events occurred in 115 patients (11.4%), with 35 (30.4%) experiencing major bleeding. Significant risk factors included venous thromboembolism (VTE) (hazard ratio [HR]: 4.003, p<0.001) and Eastern Cooperative Oncology Group Performance Status score of 1 (HR: 2.476,p< 0.001). Factors that significantly reduced hemorrhagic event risk were female sex (HR: 0.454, p=0.002) and M1a status (HR: 0.542, p=0.038). VTE is a risk factor for hemorrhagic events in patients with advanced lung cancer, and risks associated with anticoagulant therapy should be considered.” (lines 37–43)
3) The data set of the Rising-VTE/NEJ037 study was used to analyze the bleeding risk in Japanese patients with advanced lung cancer. In addition, the HAS-BLED and VTE-BLEED scores are known to be indicators of bleeding risk [22,23]. In this study, we also examined the bleeding risk using an existing scoring system. Please improve the description of this part and underline the novelty of the study.
Response: Thank you for your comment. As you pointed out, the reason for using the existing scoring was unclear; therefore, we have revised the sentence as follows:
“HAS-BLED and VTE-BLEED scores were also calculated to compare existing scoring for bleeding events and identify risk factors in this study.” (lines 68–70)
4) 2.5. Statistical analyses Data from the Rising-VTE/NEJ037 dataset were subjected to multivariate analysis using the Cox proportional hazards model with “the time to onset of a bleeding event” as the outcome. All statistical analyses were performed using SPSS Statistics version 24.0 (IBM Japan, Ltd., Tokyo, Japan). We performed multivariate analysis using the Cox proportional hazards model with the time to onset of a bleeding event as the outcome. P values <0.05 were considered to be statistically significant. Please, improve this part and underline the statistical tests used to evaluate the data.
Response: Thank you for your comment. As you pointed out, I have revised this section for clarity.
“Data from the Rising-VTE/NEJ037 dataset were subjected to multivariate analysis using the Cox proportional hazards model with “the time to onset of a bleeding event” as the outcome. All statistical analyses were performed using SPSS Statistics version 24.0 (IBM Japan, Ltd., Tokyo, Japan). We used the Cox proportional hazards model to examine the risk of bleeding in Japanese patients with advanced lung cancer. P values <0.05 were considered to be statistically significant.” (lines 129–134)
5) Figure 2 and 3. If the images can be higher resolution that would be better for the reader.
Response: Thank you for your comment. As you suggested, we have replaced Figures 2 and 3 with higher-resolution images.
6) 4. Discussion The Rising-VTE/NEJ037 study is one of the largest prospective observational studies of venous thrombosis and hemorrhagic events in Japanese patients with advanced lung cancer. To our knowledge, our study is the first to analyze the risk factors for bleeding events in patients with advanced lung cancer using large-scale data from a prospective observational study. In the Rising-VTE/NEJ037 study, the median overall survival was 24.0 months (95% confidence interval [CI]: 16.8–not estimable) in the VTE+ve and edoxaban treatment group and 19.2 months (95% CI: 16.8–21.6) in the observation (VTE –ve, no edoxaban) group, indicating no significant difference (p=0.793) [21]. This indicates that VTE treatment may contribute to improving the prognosis of patients with advanced lung cancer; it is crucial to consider the risks versus benefits of bleeding events and to administer the treatment with due caution. The discussion section needs to be improved. It could be interesting to record the aim of the study. It is necessary to be more concise in the presentation of the facts, clarifying the results obtained and comparing them with previous or similar studies. However, it is interesting to answer the questions that arise from these results, backed up by published literature.
Response: Thank you for your suggestion. Accordingly, we have re-mentioned the purpose of the research and added content that more clearly reflects the results.
“The Rising-VTE/NEJ037 study is one of the largest prospective observational studies of venous thrombosis and hemorrhagic events in Japanese patients with advanced lung cancer. Using the results of Rising-VTE/NEJ037, we performed a post-hoc analysis to evaluate risk factors for bleeding events in Japanese patients with advanced lung cancer. To our knowledge, our study is the first to analyze the risk factors for bleeding events in patients with advanced lung cancer using large-scale data from a prospective observational study. In the Rising-VTE/NEJ037 study, the median overall survival was 24.0 months (95% confidence interval [CI]: 16.8–not estimable) in the group complicated by VTE and treated with edoxaban and 19.2 months (95% CI: 16.8–21.6) in the observation group, indicating no significant difference in survival time (p=0.793) among the two groups [19]. This indicates that VTE treatment may contribute to improving the prognosis of patients with advanced lung cancer; it is crucial to consider the risks versus benefits of bleeding events and to administer the treatment with due caution.” (lines 207–219)
7) 5. Conclusions VTE is a risk factor for hemorrhagic events in advanced lung cancer, and it is necessary to consider the risks associated with the administration of anticoagulant therapy. The conclusions might be beneficial to include a sentence in the last part that briefly summarizes the key findings of the study. This can provide readers with a quick overview of the research.
Response: Thank you for your comment. As you suggested, we have mentioned in the conclusion section that VTE increases the risk of bleeding the most and that anticoagulation therapy needs to be performed after evaluating the risk of bleeding, as follows:
“In Japanese patients with advanced lung cancer, complications with VTE were observed to increase the risk of bleeding the most. Anticoagulant therapy is sometimes performed for patients with advanced lung cancer complicated by VTE; however, the indication for treatment must be determined after fully considering the risk of bleeding. In the future, we would like to consider creating a scoring system for bleeding risk when using anticoagulant therapy for patients with advanced lung cancer.” (lines 273–278)

Reviewer 2 Report
Comments and Suggestions for Authors
A very important job from a parks point of view. Asian populations are relatively underrepresented in clinical trials and the risk of both thromboembolic events and bleeding complications during treatment is high. Hence the big problem (mainly the doctor's fear of starting aggressive treatment with DOACs). At the same time, edoxaban is the last drug to join the previously existing DOACs, so there is even more need for work...
Coming back to the review, the work was nicely written, the results presented clearly (although I have a few comments below), and well discussed. As a reviewer, I have a few comments that, in my opinion, will make the work even more valuable.
- while reading the works, I could not find information on how the VTE(+)nonEdo population was treated. Please specify the treatment in the table. What medications were used? DOACs? VKA? LMWH?
- The hasbled obtained was not high. There are studies on the need to monitor concentrations that emphasize that hasbled is higher, such as therapeutic monitoring of direct oral anticoagulants—an 8-year observational study.
- Please also comment on the need to use monitored therapy. occurrence of interactions, e.g. interactions of nintedanib and oral anticoagulants, drug–food interactions of direct oral anticoagulants, or interactions with anticancer treatment. I believe that it is worth mentioning the possibility of using monitored therapy at work. the work must have educational and scientific value, it must be practical.
- education was certainly used to achieve high adherence in these patients. Please write about it. During treatment with e.g. dabigatran in patients with atrial fibrillation, adherence without adequate education is poor. What might this look like for VTE? in cancer patients?
- it is also worth noting whether there was a need to switch between therapies and what the treatment for bleeding was like. Has andexanet alfa been used? idarucizumab? PCC? FFP? Who's doing what? with what effect?
- it is worth emphasizing the practical aspects resulting from the work in the conclusions, so as to facilitate decisions about initiating this aggressive therapy in cancer patients.
Author Response
â‘ while reading the works, I could not find information on how the VTE(+)nonEdo population was treated. Please specify the treatment in the table. What medications were used? DOACs? VKA? LMWH?
Response: Thank you for your comment. As you pointed out, we agree that adding the exclusion criteria for edoxaban administration and providing information on the treatment if it was not used was necessary. Notably, in real-world setting, cases in which edoxaban could not be administered were complicated by severe renal impairment, making it difficult to administer anticoagulant therapy.
“Exclusion criteria for edoxaban administration were patients with hypersensitivity to edoxaban, decreased renal function (creatinine clearance <30 mL/min), liver disease with coagulation abnormalities, and those who have already undergone surgical treatment or treatment for recently diagnosed VTE. If the exclusion criteria for edoxaban administration were met for patients with VTE, anticoagulation therapy was administered at the discretion of the attending physician.” (lines 85–90)
â‘¡The hasbled obtained was not high. There are studies on the need to monitor concentrations that emphasize that hasbled is higher, such as therapeutic monitoring of direct oral anticoagulants—an 8-year observational study.
Please also comment on the need to use monitored therapy. occurrence of interactions, e.g. interactions of nintedanib and oral anticoagulants, drug–food interactions of direct oral anticoagulants, or interactions with anticancer treatment. I believe that it is worth mentioning the possibility of using monitored therapy at work. the work must have educational and scientific value, it must be practical.
Response: Thank you for your comment. As a companion study to the Rising-VTE/NEJ037 study, there is a study examining the pharmacokinetics of edoxaban in EGFR gene mutation-positive lung cancer patients. In the Rising-VTE/NEJ037 study, measuring edoxaban concentrations in all patients was impossible. In response to your comments, we added the following text to Limitation.
“Fifth, although there has been research on the pharmacokinetics of edoxaban in EGFR gene mutation-positive patients using some of the patients enrolled in the Rising-VTE/NEJ037 study[29], measuring pharmacokinetics and blood levels in all patients was not possible.” (lines 263–266)
â‘¢education was certainly used to achieve high adherence in these patients. Please write about it. During treatment with e.g. dabigatran in patients with atrial fibrillation, adherence without adequate education is poor. What might this look like for VTE? in cancer patients?
Response: Thank you for your comment. Unfortunately, in this study, we did not directly aim to improve patient adherence. On the other hand, as you pointed out, we agree that providing interventions to improve adherence for VTE, which requires long-term treatment, would be extremely beneficial.
â‘£it is also worth noting whether there was a need to switch between therapies and what the treatment for bleeding was like. Has andexanet alfa been used? idarucizumab? PCC? FFP? Who's doing what? with what effect?
Response: Thank you for your comment. As you pointed out, we agree that it is also important to respond when a hemorrhagic event occurs. In response to bleeding events, it is important to consider discontinuing anticoagulant therapy if it is being used or to use an antagonist, if available. In some cases, fresh frozen plasma or red blood cell transfusions may be required. In this study, bleeding events were treated by each attending physician, and there were no cases in which antagonists were used. As you pointed out, we added the following text:
“If a bleeding event occurred, treatment was performed at the discretion of each attending physician.” (lines 102–103)
⑤it is worth emphasizing the practical aspects resulting from the work in the conclusions, so as to facilitate decisions about initiating this aggressive therapy in cancer patients.
Response: Thank you for your comment. As you pointed out, we have revised the Conclusions section as shown below.
“In Japanese patients with advanced lung cancer, complications with VTE were observed to increase the risk of bleeding the most. Anticoagulant therapy is sometimes performed for patients with advanced lung cancer complicated by VTE; however, the indication for treatment must be determined after fully considering the risk of bleeding. In the future, we would like to consider creating a scoring system for bleeding risk when using anticoagulant therapy for patients with advanced lung cancer.” (lines 273–279)

Round 2
Reviewer 1 Report
Comments and Suggestions for Authors
The manuscript has been improved, as requested. No further comments.
Comments on the Quality of English Language
Minor changes of English language are required
Reviewer 2 Report
Comments and Suggestions for Authors
The authors introduced appropriate corrections to the manuscript. I believe it may be considered for publication at this time